

# Central European wind and precipitation compound events are not just due to winter storms

Miloslav Müller[1,2], Marek Kašpar[1], Vojtěch Bližňák[1]

[1]The Czech Academy of Sciences, Institute of Atmospheric Physics, Bočni II 1401, Prague, 141 00, Czechia
[2]Charles University, Faculty of Science, Albertov 6, Prague 128 00, Czechia

*Correspondence to*: Miloslav Müller (muller@ufa.cas.cz)

**Abstract.** Unlike other studies on wind-precipitation compound events, station data was employed from all seasons 1961–2020 to analyze the frequency and seasonal distribution of these events in Central Europe. The spatial pattern of the annual frequency is mainly determined by the cold half-years when the frequency generally decreases with increasing longitude (due to the decreasing effect of extratropical cyclones), but it also increases with increasing altitude (probably due to the orographic precipitation enhancement effected by strong winds). Nevertheless, wind-precipitation compound events are also generated by convective storms mainly in summer, when compound events are more equally distributed in Central Europe, with generally higher frequencies in lowlands. Five types of weather stations were distinguished according to the seasonal distribution of wind-precipitation compound events, with the percentage of summer events as the main criterion. Mostly winter type dominates in the west, mostly autumn type at the coast of the North Sea, mixed type in north-east Germany, mostly summer type in central part of Germany, and summer type in eastern part of Czechia and in south-east Austria. We also demonstrate on selected examples that compound events frequently occur at a station only in the season when both abnormal winds and abnormal precipitation events appear and are related to the same circulation conditions. This is the reason why wind-precipitation compound events are very rare at some stations, mainly in the highlands in the eastern part of the study region. We also discuss the role of the threshold for selecting wind-precipitation compound events and prove that the higher their frequency is at a station, the higher the percentage of stronger events among them. This finding highlights wind-precipitation compound events as a significant natural hazard mainly in exposed areas.

## 1 Introduction

Recently, researchers have paid attention to compound weather and climate events when a combination of multiple drivers and/or hazards contributes to societal and/or environmental risk. Thus, damage produced by a compound event can be significantly greater than the sum of the damage caused by the individual events (Arosio et al., 2020). In fact, many of the most severe weather-related and climate-related impacts are due to compound events (Zscheischler et al., 2020).

One of the frequently studied types of compound weather events are cases of simultaneous occurrence of strong winds and heavy precipitation (wind-precipitation compound events, further labeled W-P events). In coastal areas, these events can lead





to compound coastal flooding due to a combination of storm surge and high river discharges (Couasnon et al., 2020). Nevertheless, the amplifying effect of simultaneous strong winds and heavy precipitation can also apply inland, e.g., by reducing the stability of trees in highly saturated soil. The risk of increased damage to forests further increases in the case of solid precipitation as the effects of strong winds and snow loads combine.

The co-occurrence of strong winds and heavy precipitation can result from several types of storms. At the global scale, W-P
events due to tropical cyclones are most prominent (Martius et al., 2016), while extratropical cyclones are the main cause of W-P events in many parts of midlatitudes (Messmer and Simmonds, 2021). In Europe, extratropical cyclones play the most important role both in strong winds and heavy precipitation (Pfahl, 2014), but with significant regional differences in the frequency of compound events. Great Britain and Ireland are typically affected, with much higher dependence coefficients between strong winds and precipitation in their western parts (Vignotto et al., 2021). Raveh-Rubin and Wernli (2015) also
detected frequent co-occurrence of large-scale precipitation and wind gust extremes for the Mediterranean.

In the region north of the Alps, cyclones over the North Sea produce both strong winds and heavy precipitation mainly in autumn and winter, which results in a high frequency of W-P events there (Owen et al., 2021). In Eastern Europe, however, only strong winds are due to cyclones from the Atlantic (Kašpar et al., 2017), while Mediterranean cyclones are responsible for most regional precipitation extremes (Kašpar and Müller, 2010). Thus, the frequency of W-P events is considered to
decrease substantially with longitude in continental Europe (Owen et al., 2021). Nevertheless, this assumption is based on the data from the cold half-years only when severe convective storms do not typically occur in Eastern Europe. In fact, late spring and summer thunderstorms can produce both strong winds and intense precipitation in this region, as occurred, for example, in July 1984 (Kašpar et al., 2009).

Because of the significant difference between Western and Eastern Europe in terms of W-P events, the focus is on the central
part of Europe in this paper. First, the frequency of W-P events within the study region throughout the year, in both half-years and in all four seasons was analyzed. The main emphasis is given to the aforementioned zonal gradient in the frequency of W-P events and possible topographic effects because orography modifies both wind and precipitation distributions. Next, we classified the stations into five types with respect to the seasonal distribution of W-P events and explained differences among the types by differences in the seasonal distribution of strong winds and heavy precipitation and
in circulation patterns conducive to them. In the discussion, we compare two ways of evaluating W-P events in cold half-years and demonstrate how the frequency of W-P events changes with an increasing threshold for their definition.

## 2 Data and methods

### 2.1 Data

Most authors employ reanalysis data for their studies on W-P events. Because our aim is also to estimate the contribution of
convective storms, which can be very localized, we prefer measured station data, namely, daily maximum wind gusts ($F_x$) and daily precipitation totals ($P_d$), for the evaluation of strong winds and heavy precipitation, respectively.





The territories of three countries were included in this analysis, namely, Austria, Czechia, and Germany, because of their position on the boundary between Western and Eastern Europe. National weather services of the three countries (Central Institute for Meteorology and Geodynamics, Czech Hydrometeorological Institute, and German Weather Service) made their

station data available for research purposes on their websites; for technical reasons, we downloaded Austrian data via the ECAD (Klein Tank et al., 2002). To obtain a long enough study period, 60 years were considered between 1961 and 2020, although not all data series cover the whole period; however, we employed only stations with at least 50% of the data.

We also used the reanalysis data, but only when we compared the in situ detected W-P events with the large-scale circulation conditions. We have chosen the NCEP/NCAR reanalysis (Kalnay et al., 1996) for this purpose because of its coarse

resolution of 2.5°, representing large-scale processes. For each station, we used the 12 UTC values from the pixel in which the station is located. From the zonal and meridional wind components at the 850 hPa level, we determined the wind direction and speed and compared their distribution during the study period with their subsets during abnormal and compound events.

## 2.2 Definition of abnormal and compound events

For the purpose of evaluating the value extremeness, the characteristic of exceedance probability ($e$) was used. It is estimated following Eq. (1):

$$e = \frac{i}{n+1}, \tag{1}$$

where $i$ is the ranking of the value in the dataset comprising $n$ daily values. In our study, $n$ stands for all days with available data of both $F_x$ and $P_d$ regardless of the season. In the discussion part of the paper, we also use the cold half-year exceedance

probability ($e_c$), expressing the ranking of the values among $n_c$ cold half-year days; because $n_c$ is approximately half of $n$, $e_c$ is approximately twice $e$.

We are mainly concerned with extra high values with $e < 1–p$, where $p$ is a chosen threshold. We consider the 98th percentile ($p = 0.98$) as the basic threshold for defining abnormal events, which is a compromise between the need to include only truly extraordinary events and to have as large a set of them as possible. In the case of a complete data series of $n = 21915$ days at

the given station, we recognized 438 days with strong wind (called abnormal winds) and 438 days with heavy precipitation (called abnormal precipitation events). Thus, the mean annual frequency of abnormal events was 7.3 ($p = 0.98$). If we needed to distinguish events that were even more extreme, we used the 99th percentile as the threshold. Moreover, we also used other thresholds in the discussion part of the paper.

A compound event is defined by exceeding the percentile $p$ of both variables considered, with the values of $e(F_x)$ for wind

and $e(P_d)$ for precipitation less than $1–p$. The basic threshold for selecting W-P events is also $p = 0.98$.



### 2.3 Analysis of weather stations in terms of the frequency of compound events

Obviously, the number of compound events ($m$) could be between 0 and $n(1–p)$. To normalize the value, the conditional probability measure ($\chi$) was employed as the standard tool for evaluating the frequency of compound events at a station. It is estimated following Eq. (2):


$$\chi = \frac{m}{n(1-p)},$$ (2)

where $m$ is the number of days when both variables exceeded the threshold $p$ among all $n$ considered days. Thus, $\chi$ represents the percentage of compound events among abnormal winds as well as among abnormal precipitation events (Owen et al., 2021).

If wind gusts and precipitation totals were randomly distributed within the study period, the theoretical $m_{th}$ would equal $n(1–p)^2$, which is almost 9 compound events for $p = 0.98$ in the case of a complete data series of 60 years. Thus, the theoretical $\chi_{th}$ equals 2% for $p = 0.98$. To express the proportion of the actual number of compound events to the theoretical number, we introduced an additional characteristic that we call the abnormality measure $\chi^*$ and calculate it following Eq. (3):

$$\chi^* = \frac{m}{m_{th}} = \frac{\chi}{\chi_{th}}.$$ (3)

Thus, $\chi^*$ equals $50\chi$ for $p = 0.98$ and $100\chi$ for $p = 0.99$.

We are highly interested in the seasonal distribution of compound events. Thus, we also distinguish the numbers of compound events in cold and warm half-years ($m_c$ and $m_w$) and in four seasons ($m_{DJF}$, $m_{MAM}$, $m_{JJA}$, and $m_{SON}$) and calculate the respective values of $\chi_c$, $\chi_w$, $\chi_{DJF}$, $\chi_{MAM}$, $\chi_{JJA}$, $\chi_{SON}$ and $\chi^*_c$, $\chi^*_w$, $\chi^*_{DJF}$, $\chi^*_{MAM}$, $\chi^*_{JJA}$, $\chi^*_{SON}$ by (2) and (3), respectively. In this case, $n$ is replaced by actual numbers of considered days in the half-years ($n_w$, $n_c$) or seasons ($n_{DJF}$, $n_{MAM}$, $n_{JJA}$, $n_{SON}$). In the discussion part of our paper, we also determine an alternative conditional probability measure $\chi_{cc}$ based on cold half-year exceedance probability $e_c$ instead of $e$.

In contrast to $\chi$, the abnormality measure $\chi^*$ is theoretically independent of p; thus, it enables us to also discuss the role of the threshold when defining W-P events. To present the relationship between $p$ and $\chi^*$ at a station, we calculate the linear regression coefficient between $\chi^*$ and $100p$, choosing $p$ equal to 0.95, 0.96, 0.97, 0.98, and 0.99. Positive values are for increasing normalized frequency of W-P events with increasing threshold for their definition, which means an increasing risk of extra strong W-P events.


One of the main aims of the present paper is to demonstrate how longitude and altitude influence the frequency of compound events within the study region. We employed multiple linear regression for this purpose. Furthermore, we also classified the stations in terms of the seasonal distribution of W-P events. Because we recognized such events as not very common in spring, we considered only the other three seasons in this process. First, we determined the relative magnitudes of $\chi_{JJA}$, $\chi_{SON}$,

and $\chi_{DJF}$ within the sum of the three values for each station (labeled $\rho_{JJA}$, $\rho_{SON}$, and $\rho_{DJF}$, respectively). Because $\rho_{JJA}$ is the most spread between 0 and 1, we used it as the main criterion for classifying the stations into four main types, with one of





them further divided into two subtypes with respect to $\rho_{SON}$ values (Table 1). Because the sum of $\rho_{JJA}$, $\rho_{SON}$, and $\rho_{DJF}$ equals

one, we used the triangle diagram to present our results. In the diagram, the values of the relative magnitudes are plotted on a

trio of axes that are 60° to each other.

**Table 1: Types of weather stations distinguished with respect to the seasonal distribution of wind-precipitation compound events. The variables $\rho_{JJA}$ and $\rho_{SON}$ represent the percentages of these events in summer and autumn, respectively, among all events apart from spring.**

| Main type | Subtype | Abbreviation | Criteria |
|---|---|---|---|
| Winter/autumn | Mostly winter | WA | $\rho_{JJA} < 0.25$; $\rho_{SON} < 0.5$ |
| | Mostly autumn | AW | $\rho_{JJA} < 0.25$; $\rho_{SON} \geq 0.5$ |
| Mixed | | M | $\rho_{JJA} < 0.5$; $\rho_{JJA} \geq 0.25$ |
| Mostly summer | | SM | $\rho_{JJA} < 0.75$; $\rho_{JJA} \geq 0.5$ |
| Summer | | S | $\rho_{JJA} \geq 0.75$ |

## 3 Results

### 3.1 Frequency of compound wind-precipitation events

The frequency of W-P events is distributed very unequally in the study region (Fig. 1), with a maximum of $\chi = 26.6\%$

recorded in Freudenstadt (southwest Germany, in Schwarzwald Mts.) and a minimum of $\chi = 1.7\%$ in Greifswald (north-east

Germany, at the Baltic Sea). While the first value represents two W-P events per year on average, the latter value denotes

only one such event per eight years, which indicates independence between strong winds and heavy rains there. In general,

the values decrease from west to east, with local differences due to altitude and other factors. However, the Alpine region

does not fit the general rule, with maxima recorded in southeastern Austria. Considering the region north of the Alps (with

latitude > 48°N), we detect an exponential decrease in $\chi$ with longitude (coefficient of determination $r^2 = 0.33$). The linear

model also describes the relationship quite well ($r^2 = 0.26$), with $\chi$ decreasing by 2% every 3 zonal degrees. Obviously, $\chi$ also

increases with increasing altitude of the station, at least in the western part of the domain (Fig. 2a). If considering both

longitude and altitude in the multiple linear regression, $r^2$ increases to 0.38.


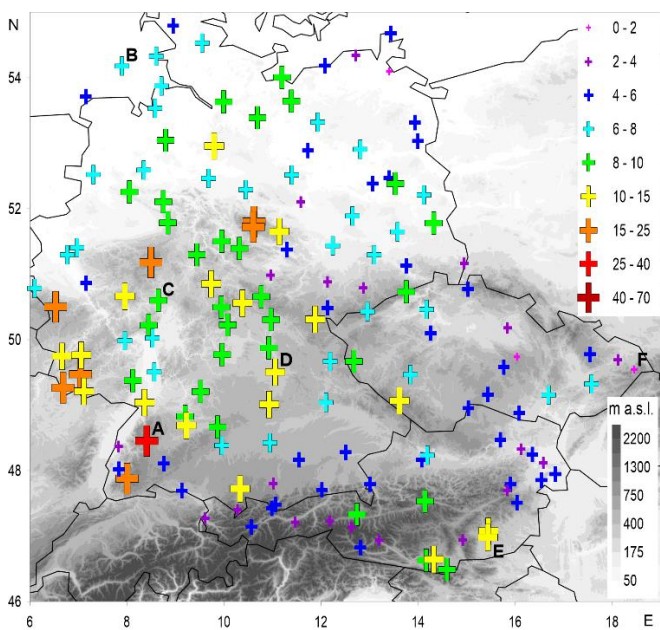

**Figure 1: Frequency of compound wind-precipitation events at individual stations throughout the years, expressed by the conditional probability measure $\chi$ [%]. For details on stations marked by letters, see Table 2.**

**Table 2: Characteristics of selected weather stations highlighted in figures, with data presented in detail in Figs. 6, 7 and 8.**

| Label | A | B | C | D | E | F |
|---|---|---|---|---|---|---|
| Station | Freudenstadt | Helgoland | Gießen | Nürnberg | Graz-Flughafen | Lysá hora Mt. |
| Latitude [N] | 48.454 | 54.175 | 50.602 | 49.503 | 46.981 | 49.546 |
| Longitude [N] | 8.409 | 7.892 | 8.644 | 11.055 | 15.440 | 18.448 |
| Altitude [m] | 797 | 4 | 203 | 314 | 340 | 1322 |
| # of days | 18202 | 21486 | 21853 | 21906 | 16338 | 15401 |
| Type | WA | AW | M | SM | S | S |
| $\chi$ [%] | 26.6 | 7.4 | 9.6 | 11.4 | 7.3 | 1.7 |
| $\chi_c$ [%] | 49.7 | 9.8 | 11.0 | 6.9 | 1.4 | 0.0 |
| $\chi_w$ [%] | 3.3 | 5.1 | 8.2 | 15.9 | 13.3 | 3.5 |
| $\chi_{DJF}$ [%] | 64.2 | 7.5 | 9.3 | 8.3 | 1.9 | 0.0 |
| $\chi_{MAM}$ [%] | 14.2 | 0.0 | 10.0 | 5.4 | 3.6 | 1.4 |
| $\chi_{JJA}$ [%] | 3.3 | 4.6 | 9.1 | 24.5 | 21.8 | 4.4 |
| $\chi_{SON}$ [%] | 25.1 | 17.8 | 10.1 | 7.3 | 1.8 | 1.2 |
| $\chi_{cc}$ [%] | 23.5 | 7.5 | 11.5 | 9.1 | 1.8 | 1.5 |
| $a$ | 2.97 | 0.20 | 0.56 | 1.12 | 1.92 | -0.12 |

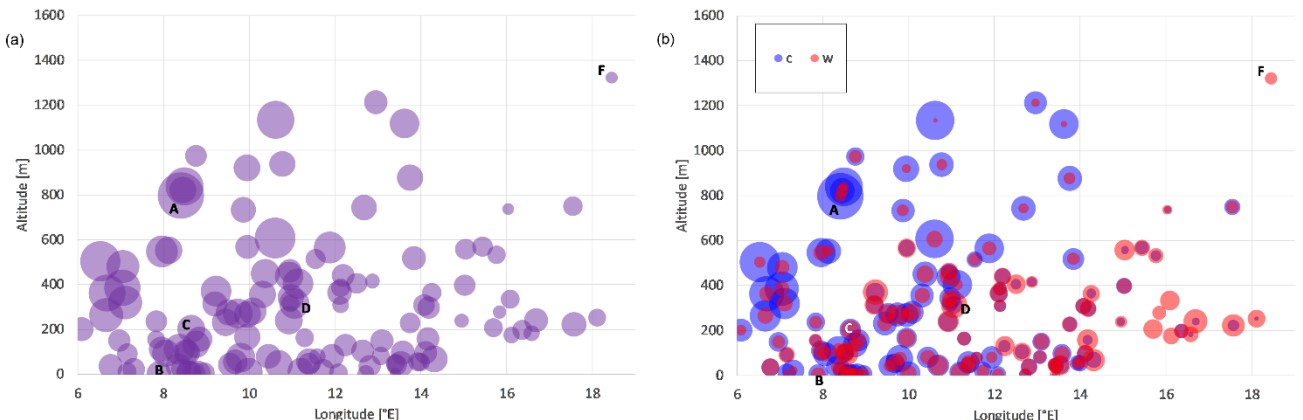

**Fig. 2: Frequency of compound wind-precipitation events at individual stations as a function of the longitude and the altitude of the site: (a) throughout the year; (b) in cold and warm half-years independently. Only stations northerly of 48°N were considered. For details on stations marked by letters, see Table 2.**

However, if we analyze cold and warm half-years separately (Fig. 3), we find that the decreasing frequency of W-P events from west to the east only fits the cold half-years and is even more pronounced than if the whole year is considered (compare

Fig. 3a with Fig. 1). Due to the decrease in $n$ to half in Eq. (2), $\chi_c$ is generally higher than $\chi$ in the western part of the domain, with maxima $\chi_c > 30\%$ at high-elevation stations. In contrast, no $\chi_c \geq 7\%$ was detected easterly from 15°E, with even zero $\chi_c$ values at some stations. Nevertheless, the proportion between the frequency of W-P events and the altitude is also more significant in cold half-years in the western part of the domain (Fig. 2b). As a result, the multiple linear regression considering both the longitude and the altitude explains almost half of the variance among stations ($r^2 = 0.49$).

The spatial pattern of W-P events is substantially different in warm half-years (Fig. 3b). While $\chi_w$ is reduced in comparison to $\chi_c$ in the west and especially in mountains (Fig. 2b), values of $\chi_w > 7\%$ frequently occur in the east of the domain, as well as in central Germany, with generally higher values in lowlands. Some low-elevation sites exhibit $\chi_w$ values over 13%, which represents the frequency of one W-P event per less than two years on average. The four highest $\chi_w$ values were recorded in Austrian lands Steyermark and Kärnten at the southeastern edge of the Alps, with the absolute maximum $\chi_w = 27.6\%$ at the

airport in Graz. Nevertheless, significantly lower values at some stations in the same region suggest large spatial variability, probably due to orographic factors.


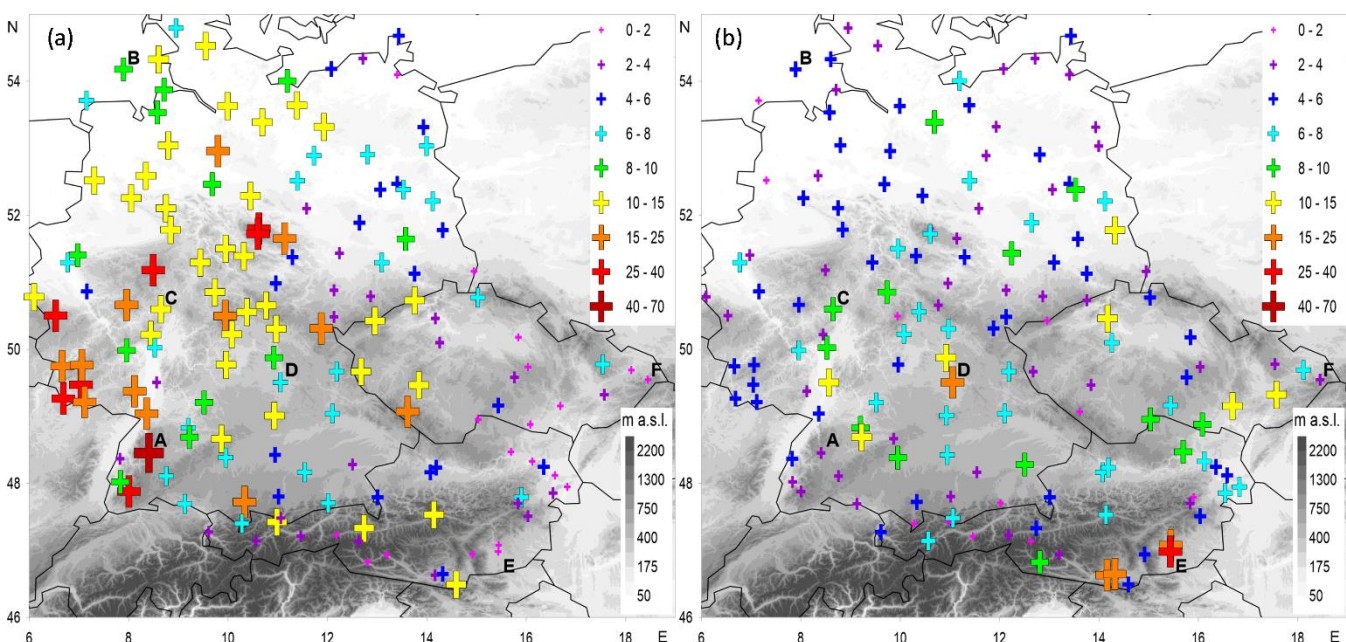

**Fig. 3: Frequency of compound wind-precipitation events at individual stations in (a) cold and (b) warm half-years, expressed by the conditional probability measures $\chi_c$ and $\chi_w$ [%], respectively. For details on stations marked by letters, see Table 2.**

The analysis of the seasonal frequencies of W-P events (Fig. 4) shows even more detailed results. In winter (Fig. 4a), the spatial distribution is very similar to that in the cold half-year (compare Fig. 4a and Fig. 3a), but the values of $\chi_{DJF}$ further increase in comparison to $\chi_c$ (in Freudenstadt, as many as almost two-thirds of winter abnormal wind and abnormal precipitation events were compounded). In contrast, compound events are generally rare in spring (Fig. 4b), which may be because March, April, and May are very different from this viewpoint; while March is similar to winter, May is reminiscent of summer, and very few W-P events appear in April in the studied region at all. Summer $\chi_{JJA}$ values are usually much higher

than $\chi_w$ (Fig. 4c), almost two times at some low-elevation locations, which suggests the crucial role of summer months within the warm half-year in terms of W-P events there. Because two of the autumn months belong to the cold half-year, one would expect that the autumn spatial distribution (Fig. 4d) is similar, but less pronounced, in comparison to the winter pattern. It fits in most of the study domain except the North Sea, where autumn W-P events are more frequent than winter

events.





**Fig. 4: Seasonal frequency of compound wind-precipitation events at individual stations in (a) winter, (b) spring, (c) summer, and (d) autumn, expressed by the conditional probability measures $\chi_{DJF}$, $\chi_{MAM}$, $\chi_{JJA}$, and $\chi_{SON}$ [%], respectively. For details on stations marked by letters, see Table 2.**






### 3.2 Classification of stations in terms of compound wind-precipitation event seasonality

The seasonal analysis of W-P event frequencies at individual weather stations enabled us to classify the stations into four main types, one of which was divided into two subtypes (Table 1, Fig. 5a). Among stations with less than 25% of summer W-P events, most stations belong to subtype WA (mostly winter), which typically occurs westerly from 10°E but is also

typical for highlands up to 15°E (Fig. 5b). All 10 stations with maximum $\chi$ belong to this type, including Freudenstadt with the highest frequency of W-P events at all. At this station, both abnormal winds and abnormal precipitation events are concentrated in the cold half-year (Fig. 6a), which makes this period very suitable for W-P events. Nevertheless, an asymmetry between the two phenomena is obvious: while 2 of 10 $F_x$ extremes were connected with abnormal precipitation, almost all $P_d$ extremes were recorded together with abnormal winds (Fig. 7a).

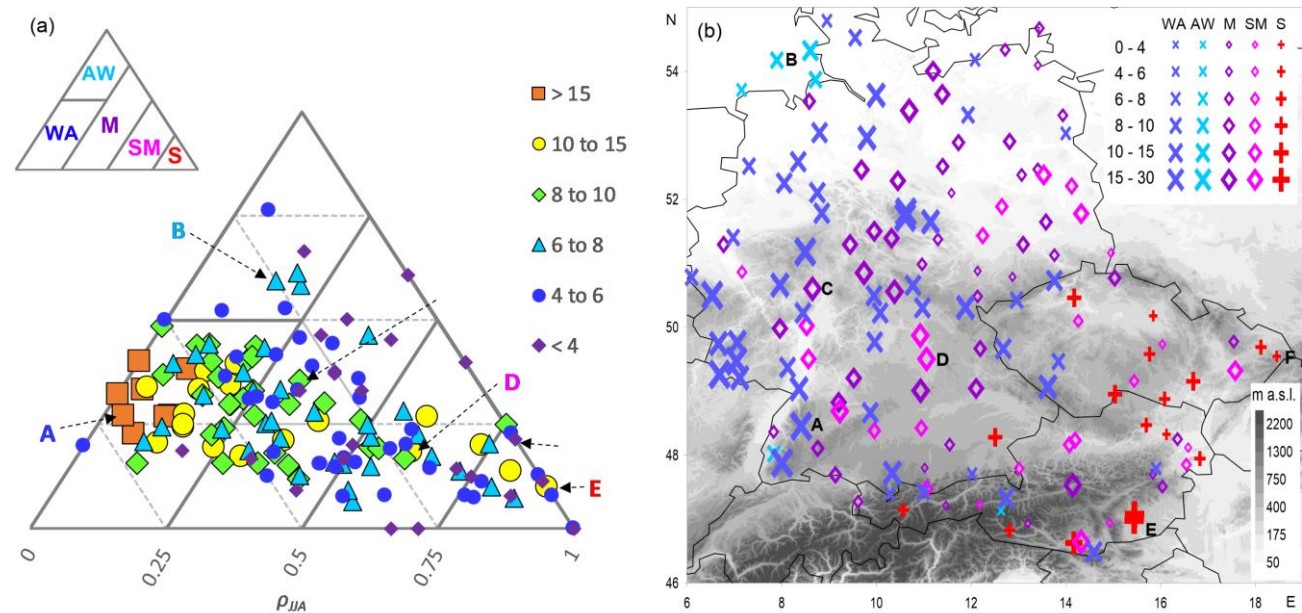


**Fig. 5: (a) Classification scheme of weather stations in terms of compound wind-precipitation events into five types. The membership of the type is given by the position of the sign, and colors are for conditional probability measures $\chi$. (b) Spatial distribution of weather stations of the five types. The size of the signs corresponds to the $\chi$ values. For details on stations marked**
**by letters, see Table 2.**




**Fig. 6: Monthly distribution [%] of abnormal daily maximum wind gusts (F$_x$) and daily precipitation totals (P$_d$) with e < 0.02 at weather stations specified in Table 2: (a) Freudenstadt, (b) Helgoland, (c) Gießen, (d) Nürnberg, (e) Graz-Flughafen, (f) Lysá hora Mt. The color part of the column belongs both to abnormal winds and to abnormal precipitation events and depicts the portion of compound wind-precipitation events among abnormal events in the month.**




**Fig. 7: Statistical distribution of daily maximum wind gusts ($F_x$) and daily precipitation totals ($P_d$) at weather stations specified in Table 2: (a) Freudenstadt, (b) Helgoland, (c) Gießen, (d) Nürnberg, (e) Graz-Flughafen, (f) Lysá hora Mt. The signs represent individual days, with colors distinguishing months; the position of the sign expresses exceedance probabilities e($F_x$) and e($P_d$). Values of the 90th, 98th(in bold), 99th, 99.9th and 99.99th percentiles of $F_x$ and $P_d$ are given for each station. The upper left, lower right, and lower left parts of the diagram contain wind-only, precipitation-only, and wind-precipitation compound events, respectively.**





Among stations with less than 25% of summer W-P events, we also recognized a subtype AW (mostly autumn) with more than 50% of W-P events in autumn. Several such stations are concentrated along the coast of the North Sea, including Helgoland Island. Fig. 6b explains why autumn is a more suitable season for W-P events than winter there: while abnormal winds are spread through the whole cold half-year, the frequency of abnormal precipitation events significantly decreases during this period. The $\chi$ values are not extremely high at these stations because summer is the typical season for $P_d$

extremes, while winter is the typical season for $F_x$ extremes (Fig. 7b).

Mixed type M consists of stations with the most equal distribution of W-P events throughout the year. Abnormal winds and precipitation events mostly occur in the cold and warm half-years, respectively, which is mainly typical for the northeastern part of Germany. Nevertheless, both phenomena are frequent enough in the opposite part of the year, and thus, W-P events are spread throughout the year (Fig. 6c). While W-P events with extreme $F_x$ values typically occur in the cold half-year,

those with $P_d$ extremes are more frequent in summer and autumn at stations of this type (Fig. 7c).

Two other types, namely, SM (mostly summer) and S (summer), are characterized by more than 50% and 75% of summer W-P events, respectively. Such stations prevail in the southeastern part of the region. At some stations in the lowland, $\chi$ can reach rather high values, with maxima of approximately one half of the maxima of the WA type. In the case of the SM type, abnormal winds are still more frequent in the cold half-year (Fig. 6d), but some of them, even extreme winds, also appear in

the opposite part of the year (Fig. 7d). The summer type S is due to a significant dominance of abnormal precipitation events in the warm half-year (Fig. 6e). In the case of low-elevation stations in southeastern Austria, $P_d$ extremes appear in late summer or even in autumn, but W-P events are concentrated only in summer because autumn $F_x$ extremes are rare (Fig. 7e).

Apart from the WA type, some stations of the other types reach very low $\chi$ values. Most of these stations are located in the Alps or in northern parts of other mountains where the annual cycles of abnormal winds and abnormal precipitation events

are completely the opposite (Fig. 6f). This fact can be explained when comparing abnormal winds and abnormal precipitation events in terms of circulation patterns conducive to them.

**3.3 Relationship between wind-precipitation compound events and circulation patterns**

The seasonal coincidence of abnormal winds and abnormal precipitation events is a necessary but not a sufficient condition for W-P events. In fact, even more important, is whether both types of abnormal weather events occur under the same

circulation conditions. To represent the circulation pattern on a given day, we decided to employ wind speed and direction at 12 UTC at the 850 hPa level, derived from a coarse resolution reanalysis (Fig. 8). In general, we can assume that if high $F_x$ values appear together with strong 850 hPa winds, they are due to deep extratropical cyclones. In contrast, high $F_x$ values without this relation are due to convective storms.



**Fig. 8: Distribution of wind at the 850 hPa level around weather stations specified in Table 2: (a) Freudenstadt, (b) Helgoland, (c) Gießen, (d) Nürnberg, (e) Graz-Flughafen, (f) Lysá hora Mt. The signs represent individual days (12 UTC); days with abnormal wind (W), abnormal precipitation (P), and compound events (W-P) in cold (c) and warm (w) half-years are highlighted by colors. Large signs are observed for compound events when both the maximum wind gust and daily total exceeded their 99th percentiles. The position of the sign with respect to the center of the diagram expresses the wind speed [ms⁻¹] and the wind direction, with westerly winds positioned in the left part of the diagram. Data are from the corresponding grid cells of the NCEP/NCAR reanalysis. For details on the presented stations, see Table 1.**



The extra high frequency of W-P events at many WA stations is because abnormal winds and abnormal precipitation events both occur when a strong 850 hPa wind blows from the western sector in the cold half-year (Fig. 8a); in the warm half-year, this pattern is rare. At AW stations with generally lower $\chi$ values, the circulation patterns of both types of abnormal weather events are much more variable, with usually strong 850 hPa winds during abnormal precipitation events shifting into the southwestern sector, which is also typical for W-P events at these stations (Fig. 8b). Thus, deep extratropical cyclones seem to be the crucial mechanism producing W-P events at stations of both subtypes.

The mixed-type M and mostly summer-type SM stations are characterized by the dichotomy of two different circulation patterns producing W-P events. The coincidence of abnormal winds and abnormal precipitation events with strong (south-)western 850 hPa winds is responsible for W-P events in the cold half-year, while much weaker southwestern 850 hPa winds are typical for W-P events in the warm half-year (Fig. 8c, d). At stations of the summer type S, W-P events are missing in the cold half-year because of fully different 850 hPa winds during abnormal winds and during abnormal precipitation events in this part of the year (Fig. 8e).

At Lysá hora Mt. and other similar stations, however, the circulation patterns of abnormal winds and abnormal precipitation events are also different in the warm half-year because high $P_d$ values are mainly recorded when the northern 850 hPa wind blows (Fig. 8f). However, this wind is usually not strong enough to produce abnormal winds at the surface, which are almost exclusively related to the southern or southwestern 850 hPa wind. Thus, W-P events are very rare there and only occur in the case of unusually strong northern 850 hPa winds due to extra deep cyclones over Eastern Europe.

# 4 Discussion

## 4.1 Uncertainty of data

The presented results could be partly influenced by the quality of the data, mainly by possible inhomogeneity in the $F_x$ series due to, e.g., changes in anemometers employed for the measurement. Thus, we tested the effect of possible additional homogenization of the data series using a statistical correction technique based on the relationship between measurements and reanalysis data (Kašpar et al., 2017). In general, $\chi$ values calculated from homogenized data do not significantly differ from those of the original data. The reason could be that possible inhomogeneity can only make small and nonsystematic changes in the ranking of $F_x$ values with respect to high $P_d$ values. On the other hand, homogenization could incorrectly reduce high $F_x$ values, mainly those produced by convective storms. Thus, we decided to use the original $F_x$ data instead of the homogenized ones in this study.

The analyzed stations differ in terms of the length of their data series and the number of possible gaps. Approximately half of the data series cover more than 80% of the study period, and the other half covers 50–80%. Data series from Austria are generally rather short because they usually start only in the 1980s. Expecting a random distribution of W-P events among the years, the respective part of the events is captured by the given station. Thus, both $m$ and $n$ in Eq. (2) decrease equally on



average, so the shorter length of the data series does not influence the analysis substantially; it only produces increased uncertainty in the results.

**4.2 Annual and half-year exceedance probability**

To enable comparison of the results with other studies that were limited to the cold half-years only, we also calculated cold half-year exceedance probability values ($e_c$) from $n_c$ cold half-year days by Eq. (1). Using $e_c$ instead of $e$, abnormal winds and abnormal precipitation events are only selected from cold half-years, and thus, the set of W-P events can also comprise weaker events from this half-year in comparison with warm half-year events. Therefore, the cold half-year conditional

probability measure $\chi_{cc}$ can significantly differ from $\chi_c$, which also evaluates the percentage of cold half-year W-P events among abnormal winds, as well as abnormal precipitation events but is selected with respect to the $F_x$ and $R_d$ distributions throughout the year.

The relationship between $\chi_{cc}$ and $\chi_c$ values for five distinguished types of stations is presented in Fig. 9. For types WA and AW with less than 25% of W-P events in summer, $\chi_{cc}$ is significantly lower than $\chi_c$ because $e_c$ is only approximately half of

$e$. In contrast, types SM and S with more than 50% of W-P events in summer overestimate $\chi_{cc}$, in comparison with $\chi_c$, because if considering only cold half-years, even rather small values of $F_x$ and $R_d$ are considered instead of warm half-year W-P events. As a result, the contrast between the western and eastern parts of the study region in terms of the cold-half year W-P frequency is much more pronounced in this study.

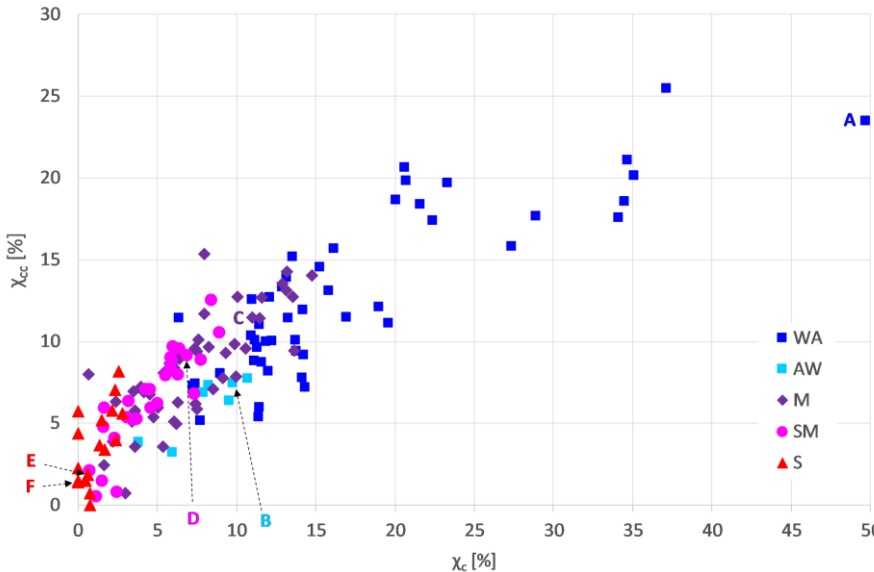

**Fig. 9: Comparison between the conditional probability measures of wind-precipitation compound events in the cold half-years, determined using the exceedance probability through the whole year ($\chi_c$) and for the cold half-years only ($\chi_{cc}$), at weather stations of five types distinguished by colors. For details on stations marked by letters, see Table 2.**

### 4.3 The threshold of abnormal and compound events

We employ the percentile $p = 0.98$ as the threshold for the definition of abnormal winds and abnormal precipitation events to
obtain enough W-P events for the analysis; the absolute values of the threshold are presented in Fig. 7 for six selected
stations. Because these values appear more than seven times per year on average, they can hardly be considered extreme.
Therefore, we also analyze trends in the frequency of W-P events by increasing the threshold for their definition from $p =$
0.95 to $p = 0.99$. We utilized the abnormality measure $\chi^*$ for this purpose because it is independent of $p$. The trend is
characterized by the linear regression coefficient $a$ between $100p$ and $\chi^*$ at individual stations.

The results of this analysis are presented in Fig. 10, which shows that the coefficient $a$ is significantly positively correlated
with the $\chi$ value both in the cold and warm half-years. At all stations with $\chi > 10\%$, $a$ is positive, and it increases with
increasing $\chi$. This fact proves that at stations with a high frequency of W-P events, the probability of extreme events is even
higher than one would expect from the presented results.

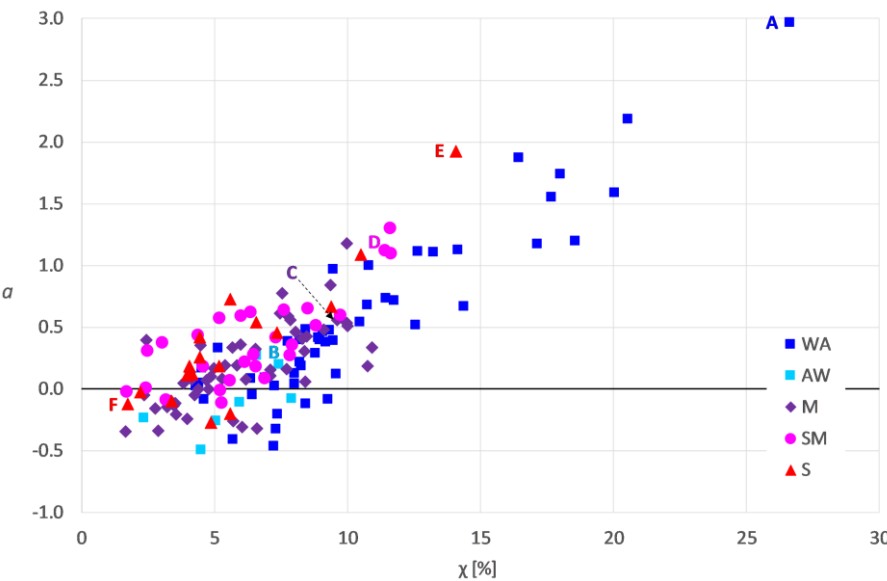

**Fig. 10: Relationship between the conditional probability measures of wind-precipitation compound events ($\chi$) and the linear
regression coefficients of the dependence of the abnormality measure $\chi^*$ on the threshold chosen for definition of the events at
weather stations of five types distinguished by colors. For details on stations marked by letters, see Table 2.**

### 5 Conclusion

Unlike other studies on W-P events in Europe, our analysis is based on station data from all years. This confirms the general
zonal gradient in the frequency of W-P events, but it also enables a more detailed view of their spatial and time distribution.
The effect of altitude is noteworthy. This increases the frequency of W-P events in the cold half-year, probably due to the
process of orographic precipitation enhancement, which intensifies with increasing wind speed. Thus, the maximum
frequency of W-P events was detected at mountain stations in western Germany, where the events are concentrated in winter.





A secondary maximum of W-P events was recognized at low-elevation stations in southeastern Austria, where almost only

summer events occur.

Another benefit of this study is the interpretation of the frequency and the seasonal distribution of W-P events at individual stations by the seasonal distribution of abnormal winds, abnormal precipitation events, and circulation conditions conducive to them. Although wind velocity at the 850 hPa level is only a simple indicator of large-scale circulation, it enabled a reasonable explanation of the W-P event distribution.

Finally, we also introduced the concept of the abnormality measure $\chi^*$, which, in contrast to the standard conditional probability measure $\chi$, is independent of the threshold chosen for the definition of W-P events. We are convinced that it can serve as an additional tool for the analysis of W-P event frequency. Our finding that the percentage of stronger events generally increases with the frequency of compound events at a station in Central Europe highlights wind-precipitation compound events as a significant natural hazard mainly in exposed areas.

**Acknowledgments**

This work was supported by the Ministry of Education, Youth and Sports of the Czech Republic [LTC19043] and the Technology Agency of the Czech Republic [SS02030040]. We also acknowledge the Central Institute for Meteorology and Geodynamics (Austria), Czech Hydrometeorological Institute, and German Weather Service for their kind provision of station weather data.

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
