# Peer review of "Central European wind and precipitation compound events are not just due to winter storms"

_Natural Hazards and Earth System Sciences, 2022_

## Author Comment (AC1)

*Dear reviewer, thank you for your work. We regret that you do not consider our article to be of sufficient quality. In our italicized answers, we try to convince you and the editor otherwise.*

The manuscript by Müller et al., aims to analyze Central European wind and precipitation "compound" events. Despite the topic is of interest, the manuscript fails in several key points:

- The authors are analyzing concurrent events and not real compound events. Please use the correct wording;

*In our opinion, the term "wind and precipitation compound event" is correctly used because strong winds and heavy precipitation not only co-occur but the damage produced by such an event can be larger than the sum of damage due to both phenomena independently. We have mentioned this fact in the second paragraph of the Introduction (there are also other examples of the possible compounding effect, e.g. rainfall damage in a house with its roof damaged by wind). Moreover, the term has already been used in many existing papers (e.g., Zscheischler et al., 2021: Evaluating the dependence structure of compound precipitation and wind speed extremes. Earth Syst. Dynam., 12, 1–16).*

- More information regarding the weather data is needed (weather stations with full data, station % in altitude, missing value, etc). Data homogeneity is still a key issue when analyzing station data. Despite the authors discussed homogenization in section 4.1, there is not enough information and the methodology points for a paper from 2017 without any further information.

*With respect to your comment, we intend to add an electronic supplement with details on all station data series, including the station altitude and time period covered. Thank you for your comment on the homogeneity of data: we make an additional homogeneity test of the data series and exclude those which may not meet its criterion.*

- I am puzzle with the use of the outdated NCEP/NCAR reanalysis. I don´t agree with the authors that the coarse resolution of 2.5°, represents well the large-scale processes. The authors should be aware that both in winter and summer, some local features (sting jets, dry intrusions, convective storms) will not be well represented in a 2.5 course resolution. The use of the state-of-the-art ERA-5 reanalysis in its native grid it´s imperative.

*Though ERA5 contains only preliminary results for first 18 years of our study period (1961–1978), we agree that it could probably be used in our study. However, we are still convinced that our decision for the coarse NCEP/NCAR reanalysis makes sense. We want to stress what we use the reanalysis data for: it serves for expressing and classifying the atmospheric circulation patterns at the synoptic (=large) scale in Central Europe, not for detailed analysis of the wind field. Each event is represented by two numbers only (zonal and meridional wind components) which need to be as general as possible to avoid effects of local factors. It is similar to the question of the appropriate horizontal resolution of a GEM: to express the position of a certain location in terms of the entire mountain range, it is not appropriate to use a detailed model, where the orientation of the slopes is influenced by local relief shapes.*

- Processes (drivers of the concurrent events) based analysis is missing from the entire manuscript. What summer and winter concurrent events differ? Which different drives take a role on different seasons?

*Due to the length of the paper, our intention was to leave the detailed analysis of the meteorological causes of the events for another, separate paper, because adding such an analysis to the existing text would approximately double its length. However, we can include several brief case studies to illustrate our findings if necessary.*

The novelty of the results is also an issue. In addition, no clear objective is clearly stated in the manuscript. I don´t believe the papers as enough novelty to be included in just a journal like the Natural Hazards and Earth System Sciences.

*We regret that obviously, we have not managed to emphasize more the goals and novelty of our research. Our objective was to determine the regularities of the spatial distribution of the frequency and the seasonality of W-P events within Central Europe. We consider the following aspects of our work and findings to be new:*

*1/ we process data for the whole year, not only the cold half-year, so unlike previous studies, we document W-P events also in areas where strong winds typically occur in the warm part of a year;*

*2/ we process station data, so we also capture local strong winds apparently related to severe convection, not captured by gridded values of maximum wind speed in reanalyses; in areas where W-P events of this kind are typical, we therefore document a much higher frequency of W-P events than previous studies;*

*3/ we consider the altitude of the stations and demonstrate that the frequency of W-P events significantly depends on the altitude but this dependence is not uniform: in the west, the frequency increases with increasing altitude, while it decreases with increasing altitude in the east of Central Europe; we also explain how this fact is related to circulation patterns;*

*4/ we demonstrate the fact that the higher the threshold of strong wind and heavy rain, the higher the relative frequency of W-P events at weather stations where W-P events are rather frequent.*

The authors fail to put the manuscript into a larger context and discuss with the main drivers. In the present form it´s just a description of the statistical results.

*We can't quite agree. If there is a high value of wind speed at 850 hPa level (yes, in the coarse reanalysis!), it proves that the event was due to a big pressure gradient within a cyclone – its position can be estimated from the 850 hPa level wind direction. On the contrary, if 850 hPa level wind was slow, the wind gust at the surface had to be due to another factor, very probably due to severe convection. Of course, this assumption could – and should – be further elaborated in our next paper.*

Therefore, I strongly suggest the rejection of the manuscript.

*We respect your opinion, even if we don't agree with it.*

*Miloslav Müller*

---

## Author Comment (AC2)

*Dear reviewer, thank you for your work and all useful recommendations. Our responses to your comments are highlighted in italics.*

Review of 'Central European wind and precipitation compound events are not just due to winter storms'

This paper investigates co-occurring wind and precipitation events in central Europe, mainly using station data. The study makes use of the conditional probability measure and performs some linear regression between this and the longitude and altitude. The stations used are also split into groups according to the time of year at which most co-occurring events take place.

Main points

Part of the motivation of the paper is stated to be the significant difference between Western and Eastern Europe. However, there is no investigation into the causes of these differences, other than showing that they have different seasonal distributions. Since the title of the paper states that winter storms are not the only cause, it would be nice to know what the other causes are.

*We concerned with the differences among the five types of stations in terms of circulation differences of W-P events in Chapter 3.3. We can reasonably assume that at stations, where we have ruled out the influence of an extratropical cyclone because of the low synoptic-scale wind speed at the 850 hPa level, the W-P events were caused by severe convection. We agree that it would be possible to further specify the causes (derecho, downbursts etc.), but this is beyond the scope of a single paper in our opinion. In fact, our intention was to leave the detailed analysis of the meteorological causes of the events for another, separate paper, because adding such an analysis to the existing text would approximately double its length.*

The introduction of the study is rather brief and does not clearly identify what the main aims or questions of the paper are. What new insight does this study hope to achieve that has not already been seen in previous work?

*We regret that obviously, we have not managed to emphasize more the goals and novelty of our research. Our objective was to determine the regularities of the spatial distribution of the frequency and the seasonality of W-P events within Central Europe. We consider the following aspects of our work and findings to be new:*

*1/ we process data for the whole year, not only the cold half-year, so unlike previous studies, we document W-P events also in areas where strong winds typically occur in the warm part of a year;*

*2/ we process station data, so we also capture local strong winds apparently related to severe convection, not captured by gridded values of maximum wind speed in reanalyses; in areas where W-P events of this kind are typical, we therefore document a much higher frequency of W-P events than previous studies;*

*3/ we consider the altitude of the stations and demonstrate that the frequency of W-P events significantly depends on the altitude but this dependence is not uniform: in the west, the frequency increases with increasing altitude, while it decreases with increasing altitude in the east of Central Europe; we also explain how this fact is related to circulation patterns;*

*4/ we demonstrate the fact that the higher the threshold of strong wind and heavy rain, the higher the relative frequency of W-P events at weather stations where W-P events are rather frequent.*

Some of the methods are not well described. For example, the exceedance probability does not seem to be relevant in the analysis - the use of the 98^th percentile threshold is all that is needed to describe what is done.

*We agree that it seems too trivial to explain exceedance probability but we needed to introduce also the cold half-year exceedance probability $e_c$. Moreover, other thresholds than 0.98 are also used in the paper and thus, we needed to mention it.*

It is not clear what the goal of the linear regressions are, or exactly how this is performed.

*We used linear regression for two main purposes: (i) to demonstrate how longitude and altitude influence the frequency of compound events within the study region; (ii) to analyze the trends in the frequency of W-P events in increasing the threshold for their definition (a sensitivity study – see another our response above). In both cases, we employed the linear least-squares method to fit a linear first-degree polynomial model to data.*

I also have reservations on the use of the NCEP/NCAR reanalysis data. Just because it has a coarse resolution, does not mean it captures the large-scale processes better than, say, a much higher resolution reanalysis.

*We understand that it looks strange to use a coarse reanalysis when a much more detailed is at disposal, though ERA5 still contains only preliminary results for first 18 years of our study period. Anyway, we are convinced that our decision for the coarse NCEP/NCAR reanalysis makes sense. We want to stress what we use the reanalysis data for: it serves for expressing and classifying the atmospheric circulation patterns at the synoptic (=large) scale in Central Europe, not for a detailed analysis of the wind field. Each event is represented by two numbers only (zonal and meridional wind components) which need to be as general as possible to avoid effects of local factors. It is similar to the question of the appropriate horizontal resolution of a GEM: to express the position of a certain location in terms of the entire mountain range, it is not appropriate to use a detailed model, where the orientation of the slopes is influenced by local relief shapes.*

The 'homogenization' of the data using the reanalysis that is mentioned in the discussion is not properly described, and so it is impossible to know if the conclusions drawn are justifiable.

*We agree with you. To support our results, we make an additional homogeneity test of the data series and exclude those which may not meet its criterion.*

The results of the study are not discussed in terms of what new insight has been added compared to previous studies. Do the results with the stations give the same chi values as previous studies? In fact the discussion and conclusion sections contain only a single reference.

*Thank you for the comment, we agree that we should not only discuss how the employed methods could influence our results but also compare our results with previous studies.*

Minor comments

1. Line 34 - a reference is needed for this statement.

*It would be fine but we are afraid that we hardly find a reference for such a general statement. In the following text, we refer to papers analyzing the relationship between strong winds and heavy rains in both tropical and extratropical cyclones and mention our case study of severe thunderstorms producing both phenomena. All these references together support the general statement.*

2. Introduction - there are other studies that have looked at co-occurring events in different seasons, just two examples are Messmer, M., I. Simmonds, 2021: Global analysis of cyclone-induced compound precipitation and wind extreme events. *Weather and Climate Extremes*, doi:10.1016/j.wace.2021.100324., and Catto, J. L., and A. J. Dowdy (2021) Understanding compound hazards from a weather system perspective, *Weather and Climate Extremes*, 32, 100313, https://doi.org/10.1016/j.wace.2021.100313.

*Thank you for your suggestion, we employ both papers to improve the introduction.*

3. Line 85 - I do not think "abnormal" is an appropriate name for the high intensity events, since they are occurring multiple times per year, one could say that they are quite normal.

*You are right, it is difficult to distinguish what is normal and what is abnormal. If you do not mind, we would like to use your term "high-intensity events" in this context.*

4. Line 98 - I'm unsure what is meant by "events among abnormal winds".

*We change the sentence as follows: "Thus, χ expresses how many percent of abnormal winds were accompanied by heavy rains (and vice versa)".*

5. Line 110 - In other words you are using the $98^{th}$ percentile calculated only from the cold season.

*Yes.*

6. Lines 116-124 - I find this paragraph rather unclear.

*Thank you, we rewrite the paragraph and explain the station classification process by mathematical formulas.*

7. Line 137 - more detail about this linear model would be useful. How is this calculated and with which stations?

*We improve the text as follows: "The linear regression model between longitude of stations and χ values…" See also one of our previous responses.*

8. Line 137 - I don't find it obvious that chi increases with increasing altitude from the figure cited.

*You are right, Fig. 1 is better for this purpose.*

9. Figure 1 - caption needs additional information of what the shading shows.

*Shading is for altitude, we add it to the caption.*

10. Table 2 - why these stations? What is the justification for looking in more detail at these specific locations?

*Five of the six stations represent five distinguished types of stations. The type S is represented by two stations in the table and figures because Graz-Flughafen is for rather specific conditions south of the Alps.*

11. Figure 2 - what is the scale for the size of the dots?

*We add the scale; it corresponds to the χ values.*

12. Line 152 - I do not understand what "the proportion between the frequency" means.

*The employ the word "relationship" instead of "proportion", thank you.*

13. Line 169 - "very different from this viewpoint" - different to what?

*Different one to each other, sorry for the unclear sentence.*

14. Line 184 - "westerly from" I think would be better as "west of".

*Ok, thank you.*

15. Figure 8 - It took me a long time to find the legend. Please put this higher up and make it larger.

*Thank you for your comment, we change the figure accordingly.*

16. Line 268 - More detail is required about this "statistical correction technique". Why do you think a reanalysis at 2.5 degrees would be good for this?

*According to one of your main points, we decided to perform a homogeneity test for each station. Thus, this paragraph will be removed.*

17. Figure 10 and the discussion of it needs more explanation. What does this mean?

*The figure supports our finding that at stations with high frequency of W-P events, their percentage even more increases with increasing the threshold. We try to explain it more clearly in the text.*

18. Conclusion - there needs to be a discussion of the results in the context of previous work.

*We agree and improve it with respect to your comment.*

*Miloslav Müller*